# Comparative Evaluation of Antioxidant Status and Mineral Composition of *Diploschistes ocellatus*, *Calvatia candida (rostk.) Hollós*, *Battarrea phalloides* and *Artemisia lerchiana* in Conditions of High Soil Salinity

**DOI:** 10.3390/plants12132530

**Published:** 2023-07-03

**Authors:** Nadezhda Golubkina, Tatiana Tolpysheva, Vladimir Lapchenko, Helene Lapchenko, Nikolay Pirogov, Viacheslav Zaitsev, Agnieszka Sękara, Alessio Tallarita, Vasile Stoleru, Otilia Cristina Murariu, Gianluca Caruso

**Affiliations:** 1Federal Scientific Vegetable Center, Kubinka 143072, Russia; 2Department of Biology, Lomonosov Moscow State University, Leninskie Gory,1, Building 1, Moscow 119234, Russia; tolpysheva@mail.ru; 3T.I. Vyazemsky Karadag Scientific Station, Nature Reserve RAS–Branch of A. O. Kovalevsky Institute of Biology of the Southern Seas of RAS, Feodosia 298188, Russia; ozon.karadag@gmail.com (V.L.); elenalapchenko@gmail.com (H.L.); 4Bogdinsko-Baskunchak Nature Reserve, Akhtubinsk 416532, Russia; npirogov2017@yandex.ru; 5Department of Hydrobiology and General Ecology, Astrakhan State Technical University, Tatisheva 16, Astrakhan 414025, Russia; viacheslav-zaitsev@yandex.ru; 6Department of Horticulture, Faculty of Biotechnology and Horticulture, University of Agriculture, 31-120 Krakow, Poland; agnieszka.sekara@urk.edu.pl; 7Department of Agricultural Sciences, University of Naples Federico II, 80055 Portici, Italy; lexvincentall@gmail.com (A.T.); gcaruso@unina.it (G.C.); 8“Ion Ionescu de la Brad” Iasi University of Life Sciences, 3 M. Sadoveanu Alley, 700440 Iasi, Romania; vstoleru@uaiasi.ro (V.S.); otiliamurariu@uaiasi.ro (O.C.M.)

**Keywords:** Bogdinsko-Baskunchak Nature Reserve, lichen, mushrooms, wormwood, mineral elements, antioxidants

## Abstract

Natural reserves play a fundamental role in maintaining flora and fauna biodiversity, but the biochemical characteristics of such ecosystems have been studied in an extremely fragmentary way. For the first time, mineral composition and antioxidant status of three systematic groups of organisms, lichens (*Diplischistes ocellatus*), mushrooms (*Calvatia candida* and *Battarrea phalloides*) and wormwood (*Artemisia lerchiana*) have been described at the territory of Bogdinsko-Baskunchak Nature Reserve (Astrakhan region, Russia), characterized by high salinity and solar radiation, and water deficiency. Through ICP-MS, it was determined that scale lichen *D. ocellatus* accumulated up to 10–15% Ca, 0.5% Fe, 15 mg kg^−1^ d.w. iodine (I), 54.5 mg kg^−1^ Cr. *Battarrea phalloides* demonstrated anomalously high concentrations of B, Cu, Fe, Mn Se, Zn, Sr and low Na levels, contrary to *Calvatia candida* mushrooms accumulating up to 10,850 mg kg^−1^ Na and only 3 mg kg^−1^ Sr. The peculiarity of *A. lerchiana* plants was the high accumulation of B (22.23 mg kg^−1^ d.w.), Mn (57.48 mg kg^−1^ d.w.), and antioxidants (total antioxidant activity: 68.6 mg GAE g^−1^ d.w.; polyphenols: 21.0 mg GAE g^−1^ d.w.; and proline: 5.45 mg g^−1^ d.w.). *Diploschistes ocellatus* and *Calvatia candida* demonstrated the lowest antioxidant status: 3.6–3.8 mg GAE g^−1^ d.w. total antioxidant activity, 1.73–2.10 mg GAE g^−1^ d.w. polyphenols and 2.0–5.3 mg g^−1^ d.w. proline. Overall, according to the elemental analysis of lichen from Baskunchak Nature Reserve compared to the Southern Crimean seashore, the vicinity of Baskunchak Salty Lake elicited increased environmental levels of Cr, Si, Li, Fe, Co, Ni and Ca.

## 1. Introduction

The Bogdinsko-Baskunchak Nature Reserve, situated in the North-Eastern part of the Astrakhan region (Russia), occupies a territory of 18,478 ha and, since 2021, belongs to UNESCO World Heritage Sites. It is famous due to the Baskunchak Salt Lake which has a sodium chloride deposit that is 10–18 m thick and is characterized by a unique semi-desert climate, strong soil salinization, remarkable gypsum accumulation, dust winds, broad annual temperature range between −40 °C in winter and up to +40 °C in summer, low precipitation (270 mm), and drought. The mentioned conditions cause significant oxidative stress to all Reserve flora and fauna species. In such extreme conditions, there exist about 507 plants [1] and 71 lichen species and about 100 families of primitive mushrooms [2].

The Reserve natural conditions provide an opportunity to study the effect of geochemical anomalies on the biochemical characteristics of Baskunchak flora representatives. In this respect, the comparison of the mineral composition and antioxidant status related to three systematic groups of flora, i.e., lichens, mushrooms and vascular plants, and the differences in metabolism intensity, life span duration and nutrition are expectedly interesting. Among the mentioned organisms, *Artemisia* species are the most attractive due to drought and salinity tolerance and the ability to accumulate significant amounts of essential oils, which are important components of natural antioxidants [3]. Among 13 identified *Artemisia* species at the Reserve, perennial *Artemisia lerchiana* Web is the most abundant, forming broad wormwood—fescue-feather grass communities [4]. Among lichens, crustose forms dominate and are mainly confined to stony substrates. The exception is represented by the epilithic lichen *Diploschistes ocellatus* (Vill.) Norman found both in limestones and soils. Lichens are poikilohydric organisms absorbing the essential nutrients for growth and development mainly from the atmosphere, and overall, the passive nutrient absorption prevails over the active absorption. Nevertheless, they are able to accumulate chemical substances also taken from the substrate, tolerating higher concentrations than higher plants and mushrooms [5]. Higher fungi belong to the third group of organisms characterized by extremely intensive growth rates of fruiting bodies during the short periods of water availability. All nutrients required for the development and growth of the mushroom fruiting body are accumulated in mycelium. No species within the three mentioned groups of organisms inhabiting the Bogdinsko-Baskunchak territory have been previously characterized in terms of the ability to accumulate macro- and trace elements, and of the intensity of antioxidant protection against unfavorable environmental conditions.

Taking into account the mentioned phenomena, the aim of the present investigation was the evaluation of mineral and natural antioxidant accumulation in wormwood *Artemisia lerchiana*, lichen *Diploschistes ocellatus* and mushrooms *Calvatia candida* and *Battarrea phalloides*, in the conditions of Bogdinsko-Baskunchak Nature Reserve.

## 2. Results and Discussion

The comparison of biochemical and mineral characteristics of lichens, mushrooms and plants grown in the same habitat has never been performed previously. To reveal peculiarities of macro- and microelements accumulation and disclose typical properties of the defense systems of the chosen organisms, the Baskunchak Nature Reserve data were compared with the results of corresponding investigations carried out at Karadag Nature Reserve, situated at the Southern shore of the Crimean Peninsula. The latter area represents the territory of the paleo-volcano and shows complex geological structure with very dry hot summers and mild winters. 

Among the organisms tested, only lichens (*Diploschistes ocellatus*), wormwood (*Artemisia lerchiana*) and *Calvatia* species were characterized previously in terms of biological activity, highlighting the interesting prospects of their utilization in pharmacology, cosmetics and medicine (Table 1). Up to date, no data regarding the biological activity of *Battarrea phalloides* (sandy stiltball) have been reported. Indeed, the only information available is devoted to the description and geographical distribution of *Battarrea phalloides*. Despite the wide habitat, including European countries, America, Africa and Australia, the mentioned mushroom may be found only episodically, which suggests the need for the special protection of this species [6,7,8]. 

### 2.1. Mineral Composition

Biological activity and level of tolerance to salinity, drought and high insulation in plants, lichens and higher mushrooms are determined by their physiological peculiarities, specific accumulation of biologically active compounds and mineral composition, which reflects both genetic features of the species and environmental biogeochemical characteristics [12,13,14]. In this respect, the mineral composition of *Diploschistes ocellatus*, *Calvatia candida*, *Battarrea phalloides* and *Artemisia lerchiana* represents a valuable parameter to compare lichens, mushrooms and plants in terms of adaptability to stress conditions (Table 2). 

Notably, among the above-enumerated organisms, only *D. ocellatus* has been described as a natural hyperaccumulator of Ca [15], while up to date no information exists about the total elemental profile of this lichen and other species investigated. The comparative evaluation of *Diploschistes ocellatus*, *Calvatia candida*, *Battarrea phalloides* and *Artemisia lerchiana* biochemical and mineral composition indicates specific peculiarities of these organisms characterized by different nutrition features in conditions of intensive oxidative stress.

Ca-hyperaccumulation is typical in *D. ocellatus* and is recorded at different biogeochemical conditions: Baskunchak Nature Reserve and the Southern shore of Crimea (Ca concentrations reached 11.5% and 10.0%, respectively), despite significant differences in the availability of other elements (Figure 1).

Table 2 data indicate that Ca content in *Artemisia lerchiana*, *Calvatia candida* and *Battarrea phalloides* are 11.8 and 31.9 times lower, respectively, than Ca level in lichen. Lichen Ca coating in the form of Ca oxalate crystals in dry habitats can be facilitated by the abundance of Ca in the substrate [16] and is supposed to contribute to water conservation in thalli [17,18]. In this respect, Ca hyperaccumulation in *D. ocellatus* may be considered as a specific form of protection against active oxygen forms and UV radiation excess [19,20,21]. 

In living organisms, Ca is accompanied by Sr and there is a close relationship between these elements showing similar ionic radius and charge (Figure 2).

As shown in Figure 2, Ca/Sr ratio changes from 1289 in *Calvatia candida* to 1005 in lichen, and 179–202 in *Battarrea phalloides* and *Artemisia lerchiana.* In this respect, both absolute Ca and Sr levels (Table 2), and Ca/Sr ratios for puffball (1) and stiltball (3) (Figure 2), reveal great interspecies variability between these mushrooms.

On the contrary, the accumulation of other macro-elements (K, Mg, and P) was the lowest in *D. ocellatus* receiving nutrients mainly through the symbiosis with *Trebouxia Puymaly* [10]. Low accumulation of Na seems to be typical in *D. ocellatus*, residing in highly salty area (EC 105 mS m^−1^) while low Na concentration in *Battarrea phalloides* presumably reflects decreased levels of total dissolved solids in the soil at the Green Garden area (EC 25 mS m^−1^) (Figure 3A). Taking into account that *Calvatia candida* and *Battarrea phalloides* are saprotrophic organisms, they accumulate the highest levels of P, ranging from 4000 to 4366 mg kg^−1^ d.w. 

Potassium and Mg were at close levels in *A. lerchiana* and the fruiting body of *Calvatia candida*, with the lowest concentration in lichen, which indirectly indicates similar nutrient sources for the former organisms compared to lichen. Contrary, among the species studied, *Battarrea phalloides* was the leader of Mg accumulation in the conditions of the Bogdinsko-Baskunchak Nature Reserve.

The K/Na ratio in plant tissues is often used as an indicator of an organism’s ability to maintain osmotic pressure in cells under salt uptake. This ratio was the highest in the *Battarrea phalloides* fruiting body (24.5), reached 6.6 in *A. lerchiana* and did not exceed two in *Calvatia candida* and *D. ocellatus*. *A. lerchiana* K/Na ratio in the vicinity of Ellton Salt Lake, in the neighboring Volgograd region, was 2.8 [22], which indicates the existence of some similar peculiarities in both salty regions. *A. lerchiana* is known to maintain high water levels in tissues under salt supply up to 800 мM NaCl. Its photosynthetic apparatus tolerance to insulation excess and low water potential in soil and in salty conditions has been shown to relate to the peculiarities of K and Na accumulation [4].

The comparison of the mineral composition of the organisms investigated revealed that the highest levels of Fe, Mn, Li, I and Si were recorded in *D. ocellatus*, while *Battarrea phalloides* accumulated predominantly B, Cu, Fe, Mn, Se and Zn (Figure 3B). 

The ability of certain mushroom species to accumulate high levels of Se, Zn and Cu may protect them against oxidative stress caused by climate factors and heavy metals [23,24]. Mineral analysis revealed 3.8 times higher Fe concentrations in *D. ocellatus* at Baskunchak, compared to the corresponding data of the Crimean lichen. This phenomenon is in accordance with high Fe content in the Baskunchak environment, reflected in the well-known presence of red water in the Gorkaya River which flows into the lake. 

At the same time, highly significant Fe accumulation in the fruiting body of *Battarea phalloides* and *Artemisia lerchiana*, compared to lichen, should be highlighted (Table 2, Figure 3B). Iron content in lichen is 6–35 times higher than in mushrooms and 11 times higher than in *A. lerchiana.* Higher levels of certain metals in *D. ocellatus*, compared to *A. lerchiana* and mushrooms (Figure 3C)*,* relate to the ability of epigeic lichens, grown in substrates with high metals content, to accumulate them at higher concentration than in the substrate [25,26,27], and the precipitation phenomenon of dust containing lithospheric elements, such as Fe at the thallus surface, with their subsequent assimilation. Fe accumulation in fruiting bodies of *Battarrea phalloides* and *Calvatia candida* is a good example of wide species differences in the ability to accumulate Fe.

Compared to Crimea, at Bogdinsko-Baskunchak Nature Reserve higher accumulation of iodine (by 12 times), Li (3.6 times), Co (4.3 times) and Cu (almost twice) was recorded in *Diploschistes ocellatus*. The detected differences in elements accumulation reflect biogeochemical anomalies of the regions investigated and lichen genetic peculiarities. At the same time, the phenomenon of Cu hyperaccumulation in *Battarea phalloides* has been recorded for the first time in the present work (Table 2). In general, the latter mushroom species proved to be a powerful hyperaccumulator not only of Cu, but of B, Mn, Zn and Se. The latter phenomenon is of special importance because high soil salinity (such as that at the territory of Bogdinsko-Baskunchak Nature Reserve) reportedly inhibits the accumulation of Se [28].

Antioxidant properties of Se may be highly valuable for *Calvatia candida* and *Battarrea phalloides* grown in conditions of oxidant stress. The ability of Se hyperaccumulation is typical only of several mushroom species [24]. The present results indicate that *Calvatia candida* and *Battarrea phalloides* at the territory of Baskunchak Nature Reserve accumulated 2–25 times higher Se than lichen and 7–77 times higher than *A. lerchiana* (Table 2). Interestingly, high levels of Se in the *Battarrea phalloides* fruiting body were recorded both at the Bogdinsko-Baskunchak Nature Reserve in Astrakhan region and at Karadag in Crimea.

At the same time, *A. lerchiana* demonstrated increased levels of B and Mn compared to *D. ocellatus* and *Calvatia candida* data. In the conditions of Baskunchak Nature Reserve, Fe/Mn ratio varied from 174 in *D. ocellatus* to 16 and 13.4 in the fruiting bodies of *Calvatia candida* and *Battarrea phalloides*, respectively, and 7.7 in *A. lerchiana*. Mn is an essential element for all living organisms, participating in photosynthesis, respiration, free radicals scavenging, protection from pathogens and hormonal signaling [29]. 

Regardless of the growing area, lichen is characterized by high levels of Ca, Fe, Al, Mg, which relates either to the geochemical peculiarities of the two regions investigated or to these elements’ assimilation in thallus consequent to their air transfer from the soil surface through dust. Indeed, dust storms are typical in summer in the Astrakhan region. High levels of Al and Mg in lichens were recorded in regions rich with dolomites and lime substrates [30]. Significant amount of metals causing damage to vascular plants, is present in lichens in non-soluble forms (as oxalates), which results in physiological deactivation of these metals [31]. Indeed, Fe is deposited in the cell wall intercellular spaces of mycobiont [25].

The comparison between lichen mineral composition at Bogdinsko-Baskunchak Nature Reserve and at the Southern Crimean shore indirectly arose the insignificant differences in As, Ca, K, Na, P, B, Mo, Se accumulation reflecting both similarities of biochemical characteristics and genetic peculiarities of the species (Figure 2). The accumulation levels of mineral elements in different lichens may significantly differ and this characteristic may be used for species distinction [32]. On the other hand, at the territory of the Baskunchak Nature Reserve the high concentration of Al, Cd, Cr, Co, Fe, I, Li in lichens is predominantly determined by wind, as it is known that except for the substrate, additional Al and Fe source relates to dust precipitation on lichen thallus [33]. 

Plants are able to accumulate heavy metal cations, absorbing them also from air and atmospheric precipitations, but the highest accumulation ability is typical of lichens. In conditions of background heavy metal concentrations in air, their mean concentrations in lichens are usually higher than in the ground part of higher plants [34]. The comparison between *A. lerchiana* mineral composition from the Baskunchak Nature Reserve and the Karadag paleo-volcano territory highlights the preferential accumulation of most elements in Karadag plants, which reflects the long-term effect of the volcanic activity at Karadag (Figure 4). Indeed, high levels of Li, B and Se in the environment as a result of volcanic activity were recorded [35,36], even at Karadag Nature Reserve [37]. On the contrary, Cd, Pb and Sr levels in *A. lerchiana* in the vicinity of Baskunchak Lake were significantly higher than the corresponding values recorded in Karadag plants.

Overall, among the species studied, *D. ocellatus* was the leader in the accumulation of Al, As and the heavy metals Cr, Ni, Pb, Sr and V. The ability of Pb hyperaccumulation in *Diploschistes* species without showing harmful effect relates to cell protection mechanism from the toxic effect of this element. Due to increased synthesis of oxalates, Pb precipitates as insoluble salts which stimulates its high accumulation and exclusion from the metabolism, as previously shown in *Diploschistes muscorum* [38,39]. Among the mushrooms investigated, *Battarrea phalloides* demonstrated an unusually high ability to accumulate Sr, Ni, Cr, As and V. 

### 2.2. Antioxidants

Stress is known to stimulate the formation of active oxygen forms and promote the biosynthesis of natural antioxidants. Among the latter, polyphenols occupy the leading position, indicating to a large extent the adaptive capacity of plants, lichens, and mushrooms [4,40,41,42]. Difficulties in the evaluation of the total antioxidant activity and polyphenol content include the choice of methods allowing for the maximum extraction of active components, significant effect of biogeochemical conditions and plant phenological phase. All these factors represent issues to manage for obtaining optimal results and their comparison with the published data. The results of the present investigation related to *D. ocellatus* demonstrated that the utilization of 70% ethanol using short-term heating is the most effective method for polyphenol (TP) extraction (Figure 5). 

The efficiency of this extraction is determined by the short duration, quick deactivation of enzymes at 80 °C, which prevents polyphenols destruction, stable at high temperature [43], and low toxicity of ethanol compared to methanol. In this respect, it is interesting to indicate that polyphenol content in *Calvatia candida* recorded in the present work was 10 times higher than the previously reported values in conditions of prolonged extraction (24 h) at room temperature [41], though such differences are also connected with different levels of oxidative stress. 

The results obtained in selected conditions indicate that *A. lerchiana* is characterized by the highest total antioxidant activity (AOA) and polyphenol content (Table 3).

Indeed, *A. lerchiana* AOA was 32 times higher than the AOA of lichen and mushrooms, while *A. lerchiana* polyphenol content was double compared to the values detected in *Calvatia candida* and *D. ocellatus*. Despite the different environmental conditions, these parameters did not significantly differ within both *D. ocellatus* samples gathered at the Baskunchak Nature Reserve and the Southern Crimean shore and *A. lerchiana*. The high values of *A. lerchiana* AOA may be also connected with high levels of essential oils [3,4]. Notably, in the list of the investigated organisms, *Battarrea phalloides* demonstrated the lowest antioxidant status, which was partly compensated by unusually high levels of Se accumulation. Indeed, Se is known to be another natural antioxidant capable of protecting plants from different forms of oxidant stress, caused by biotic and abiotic factors [12].

This trace element is not essential for plants, though its accumulation levels are determined both by biogeochemical peculiarities of the territory and genetic characteristics [44]. In his review, Falandysz [24] indicated several hyperaccumulators of Se among edible and toxic mushrooms (Boletus, Amanita, etc). The ability of *Calvatia candida* to accumulate high Se levels has also been demonstrated in this work. On the contrary, we have recorded for the first time a powerful Se hyperaccumulation in the *Battarrea phalloides* fruiting body. Se concentrations in *D. ocellatus* and *A. lerchiana* did not significantly differ between Baskunchak and Crimea, which indirectly indicates similar bio-accessibility of soil Se and aerosol transfer of this element. Nevertheless, Se soil bio-accessibility at Baskunchak Nature Reserve needs special investigations due to extremely high salinity predominantly caused by magnesium chloride (90% MgCl_2_, 9% MgSO_4_, 1% NaCl), gypsum deposits and water deficiency. Sulfur as a Se analog is known to prevent Se accumulation in plants [28,45].

### 2.3. Proline

The amino acid proline is essential for plants under stress conditions. In addition to being an excellent osmoregulator, proline also acts as a metal chelator, antioxidant and signaling molecule. In stress conditions, proline biosynthesis is enhanced to maintain osmotic balance. It stabilizes membranes, avoiding electrolytes leakage, normalizing the levels of oxygen active forms thus preventing oxidant stress. Proline supplementation is known to increase plant tolerance to stress factors and stabilize subcellular structures (membranes and proteins) via scavenging free radicals. In many plants, proline accumulation level correlates with stress resistance [46,47]. 

Table 3 data indicate that at the territory of Bogdinsko-Baskunchak Nature Reserve, *A. lerchiana* accumulated five times more proline than *D. ocellatus* and mushrooms. This phenomenon is determined to a large extent by the peculiarities of the organisms’ development: *A. lerchiana* is characterized by long vegetation period and high salinity tolerance; *D. ocellatus* spends most time in quiescence, while the fruiting body of mushrooms is ephemeric and has an extremely short life. Close values of proline accumulation at Baskunchak Nature Reserve and the Southern Crimean shore indicate similarities in oxidant stress values at these territories.

### 2.4. Malonic dialdehyde (MDA)

Metabolic differences in proline accumulation by *Diploschistes ocellatus*, *Calvatia candida*, *Battarrea phalloides* and *A. lerchiana* are also connected with malonic dialdehyde (MDA) accumulation, reflecting the intensity of lipid peroxidation and the levels of oxidant stress (Figure 6). Indeed, the highest MDA levels were recorded in mushrooms, with the highest rate of metabolism and extremely limited duration of life cycle, and the lowest in *D. ocellatus*, which may relate, priority, to the possibility of development of physiological processes in wet thallus and its complete cessation in dry thallus. Taking into account that mostly lichens are quiescent, the effect of stress factors is negligible for them. MDA level in Baskunchak *Artemisia lerchiana* was close to that of plants grown in the vicinity of Elton Lake (Volgograd region) [22] and at the territory of Karadag Nature Reserve (Figure 6). Products of lipid peroxidation are known to change ionic channels, including calcium [48].

### 2.5. Carbohydrates

Monosaccharides belong to another group of osmoregulators in plants. These compounds are practically absent in the fruiting bodies of mushrooms and lichen. On the contrary, *A. lerchiana* was shown to accumulate up to 3.6% of monosaccharides per d.w., which indicates the high adaptability level of this plant. In the conditions of Elton Lake vicinity, *A. lerchiana* accumulated only 2% monosaccharides [22]. 

### 2.6. Photosynthetic Pigments

Table 4 data provide the first information regarding chlorophyll a, b and carotene content in *D. ocellatus* and *A. lerchiana* at Bogdinsko-Baskunchak Nature Reserve. Chlorophyll content in *A. lerchiana* was 6.6 times higher than in lichen, while carotene content was almost 30 time lower. The results of the present investigation are in accordance with Shmakova and Markovskaya report [49], who investigated plants and lichens at the territory of Spitsbergen, showing the increase in chlorophyll and carotene content as well as chlorophyll/carotene ratio, with the improvement of organism organization with relative stability of chlorophyll a/b ratio. The latter parameter did not significantly differ between *D. ocellatus* and *A. lerchiana* (2.2), while chlorophyll/carotene ratio reached 7 and 30.7 in *A. lechriana* and *D. ocellatus*, respectively. Notably, in the conditions of Bogdinsko-Baskunchak Nature Reserve, *A. lerchiana* accumulated predominantly chlorophyll b, while in the vicinity of Elton Lake chlorophyll a prevailed in wormwood leaves [22].

Lichens are symbiotrophic organisms, whereas photosynthesis is achieved by algae and/or cyanobacteria. In laboratory experiments, the photobiont *Trebouxia* sp. TR9 under salinity stress conditions decreased chlorophyll content with salt concentration increase [50]. As lichens are poikilohydric organisms, high photosynthesis efficiency may take place only in conditions of optimal thallus water saturation [51]. *Diploschistes* species inhabit arid habitats [20], thus their metabolic activity covers a very short time span, and active metabolism processes coincide with the rainy season. Laboratory experiments with *D. diacapsis* from arid habitats revealed a close relationship between photosynthesis, respiration intensity and chlorophyll content in water saturated thallus [52]. While many lichen species demonstrate photosynthesis decrease in conditions of high thallus water saturation, *D. muscorum* photosynthesis rate remains close to the maximum even in conditions of extremely high thallus water saturation [53]. *D. scruposus* demonstrates not significant photosynthesis depression in the latter conditions [53]. It is supposed that such reaction of *Diploschistes* representatives relates to the presence of hydrophobic proteins in micobiont and interactions between the symbionts [53,54]. 

The scant data regarding photosynthetic pigment accumulation in lichens and wormwoods in conditions of intensive oxidant stress, such as soil salinity, entails the necessity of further investigations at the territory of Bogdinsko-Baskunchak Nature Reserve.

## 3. Material and Methods 

Twelve samples of lichen *Diploschistes ocellatus*, nine of puffball *Calvatia candida* fruiting bodies, 12 of *Battarrea phalloides* and 18 of wormwood *Artemisia lerchiana* were gathered at the Bogdinsko-Baskunchak Nature Reserve territory in Russia (Figure 7A) in May 2021 and 2022. Lichen thalli were collected at the soil surface. 

The Crimean samples of *Artemisia lerchiana* (Karadag Nature Reserve), *Battarrea phalloides*, *Calvatia candida* (Karadag Nature Reserve), and *Diploschistes ocellatus* (Novy Svet settlement) were compared in this investigation (Figure 7B).

Gathered samples were ground and dried at room temperature and homogenized.

### 3.1. Mineral Composition

The determination of 25 elements (Al, As, B, Ca, Cd, Co, Cr, Cu, Fe, I, K, Li, Mg, Mn, Mo, Na, Ni, P, Pb, Se, Si, Sn, Sr, V, and Zn) was performed on dried homogenized samples using ICP-MS on quadruple mass-spectrometer Nexion 300D (Perkin Elmer Inc., Shelton, CT, USA), according to Golubkina et al. [55].

### 3.2. Total Polyphenols (TP)

Total polyphenols were assessed in 70% ethanol extract using the Folin–Ciocalteu reagent [43]. The extraction of antioxidants was done on dry homogenized samples at 80 °C for 1 h. After cooling, adjusting the volume to 25 mL, and filtrating, 1 mL of the resulting solution was mixed with 2.5 mL of 20% Na_2_CO_3_ solution and 0.25 mL of diluted (1:1) Folin–Ciocalteu reagent in a 25 mL volumetric flask, and the volume was brought to 25 mL with distilled water. One hour later, polyphenol content was determined based on the absorption of the reaction mixture at 730 nm. As an external standard, 0.02% gallic acid was used. The results were expressed as mg of Gallic Acid Equivalent per g of dry weight (mg GAE g^−1^ d.w).

### 3.3. Antioxidant Activity (AOA) 

The antioxidant activity of samples was determined using a redox titration of 0.01 N KMnO_4_ solution with ethanolic extracts of dry samples, produced as described in the Section 3.2 [43]. Gallic acid was used as an external standard. The values were expressed in mg Gallic Acid Equivalents (mg GAE g^−1^ d.w.).

### 3.4. Proline

Proline concentration was determined according to Quertani et al. [46], with slight modifications. About 50 mg of dry homogenized *Artemisia* leaves or 100 mg of dried lichen/mushrooms powder were grinded with 10 mL of 3% sulfur salicylic acid in a mortar. After filtration, 1 mL of the resulting filtrate was mixed with 2 mL of ninhydrin reagent and 2 mL of acetic acid and then heated at 95 °C for 1 h. Proline concentration was assessed using the absorption value of the reaction mixture at 505 nm and a calibration curve with five different proline (Kenilworth, NJ, USA, Merck) concentrations.

### 3.5. Malonic Dialdehyde (MDA)

Lipid peroxidation was measured by tracing malonic dialdehyde content using thiobarbituric acid method as described by Heath and Parker [56], with a small modification. About 0.1 g of dried sample were mixed with 10 mL of 0.5% thiobarbituric acid solution and heated at 95 °C for half an hour. After cooling, the mixture was filtered and the filtrate was subjected to measurements by a spectrophotometer Unico 2804 UV (Suite E Dayton, NJ, USA). MDA content was calculated using the absorption value at 232 nm and the extinction value of 155 mM cm^−1^.

### 3.6. Carbohydrates

The monosaccharides were determined using the ferricyanide colorimetric method, based on the reaction of potassium ferricyanide with monosaccharides and fructose as an external standard [57]. The total sugars were analogically assessed after acidic hydrolysis of water extracts with 20% hydrochloric acid. The results were expressed in % per d.w.

### 3.7. Photosynthetic Pigments

Photosynthetic pigments accumulation was evaluated by spectrophotometer (Unico 2804 UV, USA) using 96% ethanolic extracts of dry samples according to Lichtenthaler [58]. The following equations were used to calculate chlorophyll and carotene concentrations:Ch-a = 13.36 A_664_ − 5.19 A_649_;
Ch-b = 27.43 A_649_ − 8.12 A_664_;
C c = (1000 A_470_ − 2.13 Ch-a − 87.63 Ch-b)/209;
where A = Absorbance, Ch-a = Chlorophyll a, Ch-b = Chlorophyll b and
C c = Carotene.

### 3.8. Statistical Analysis 

The data were processed by the analysis of variance and mean separations were performed through the Duncan’s multiple range test, with reference to 0.05 probability level using SPSS software version 27. 

## 4. Conclusions

The results of the first biochemical and mineral characterization ever performed of organisms from different systematic groups, such as lichens, mushrooms and plants, in conditions of increased salinity at the Baskunchak Nature Reserve (Russia) revealed the peculiarities of mineral composition and antioxidant status. *Diploschistes ocellatus* displayed hyperaccumulation of Ca, Al, Cr, Sr, Fe, Si, with minimum levels of K, Na, P, Cd, Cu and Zn; in the fruiting body of *Calvatia candida*, intensive accumulation of Se, Na, P and Cd was recorded and extremely low levels of Al, As, Cr, Ni, Pb, V, Co, Li and Mo, and the highest MDA accumulation; in the fruiting body of *Battarrea phalloides*, hyperaccumulation of Fe, Mn, Se, Zn and Sr and the lowest accumulation of Na was found; in *A. lerchiana,* the highest antioxidant activity, proline and monosaccharides content and high Mn and B levels were detected. Further investigations are needed to study plant responses to intensive oxidant stress in the conditions of the Baskunchak Nature Reserve.

## Figures and Tables

**Figure 1 plants-12-02530-f001:**
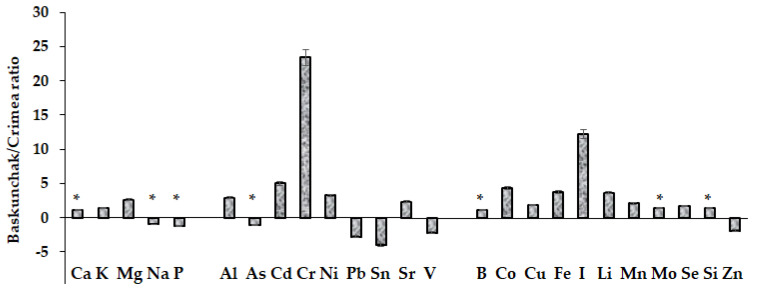
Differences in macro- and microelements accumulation in lichen *Diploschistes ocellatus* at the territory of Baskunchak Nature Reserve and the Southern Crimean shore; (*)—lack of statistically significant differences between Baskunchak/Crimea data.

**Figure 2 plants-12-02530-f002:**
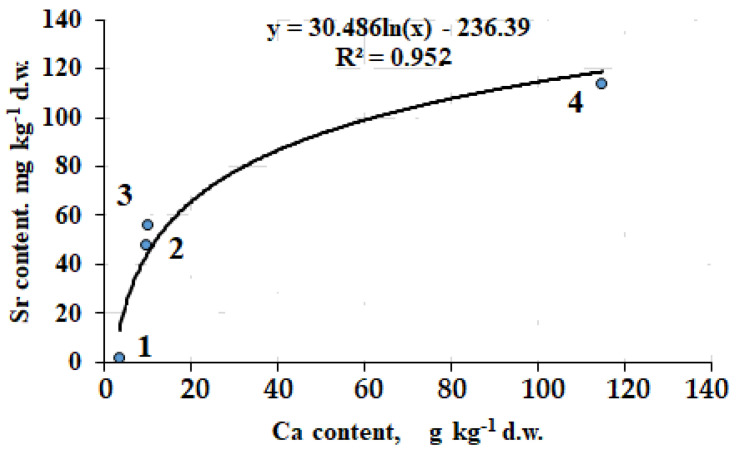
Correlation between Ca and Sr accumulation in *Calvatia candida* (1), *Artemisia lerchiana* (2) *Battarrea phalloides* (3) and *Diploschistes ocellatus* (4) at the Bogdinsko-Baskunchan Nature Reserve.

**Figure 3 plants-12-02530-f003:**
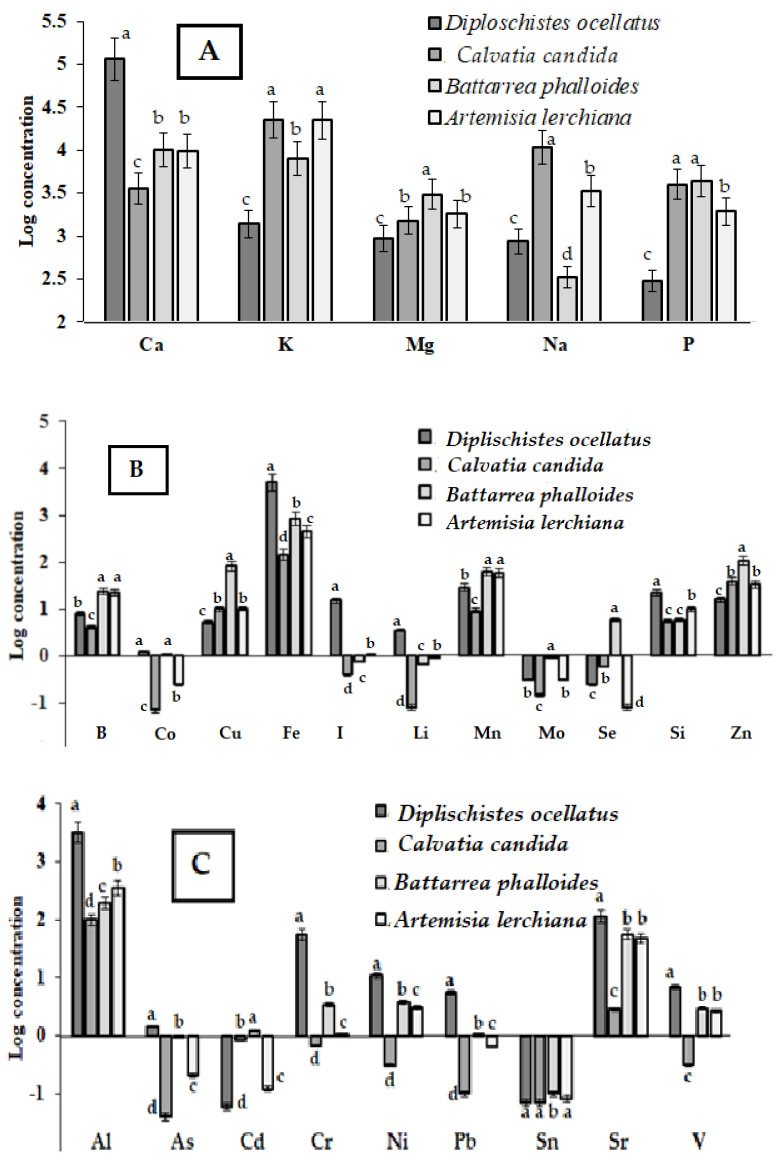
Accumulation of macro- (**A**), microelements (**B**), and As, Al and heavy metals (**C**) in lichen *Diploschistes ocellatus*, mushrooms *Calvatia candida* and *Battarrea phylloides*, and plant *Artemisia lerchiana* at the territory of Bogdinsko-Baskunchak Nature Reserve. For each element, values with the same letters do not differ statistically according to Duncan test at *p* < 0.05.

**Figure 4 plants-12-02530-f004:**
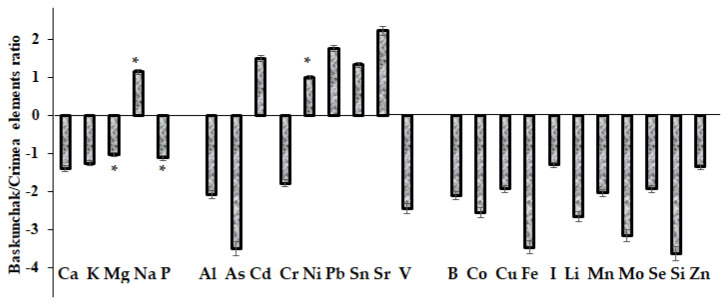
Differences in macro- and microelement accumulation in *Artemisia lerchiana* at Baskunchak and Karadag Nature Reserves (Crimea). (*): lack of statistically significant differences between Baskunchak/Crimea data.

**Figure 5 plants-12-02530-f005:**
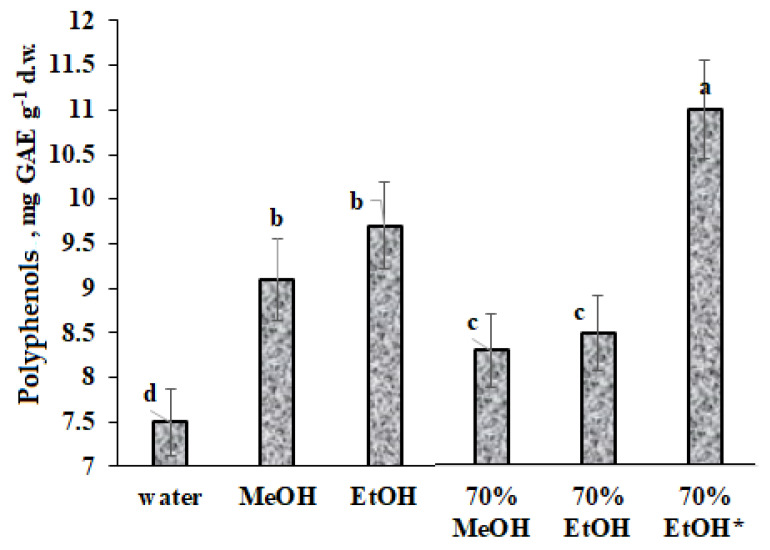
Effect of extraction conditions on polyphenol (TP) content in *Diploschistes ocellatus* (*—80 °C, 1 h; in other cases, 60 °C, 1 h). Values with the same letters do not differ statistically according to Duncan test at *p* < 0.05.

**Figure 6 plants-12-02530-f006:**
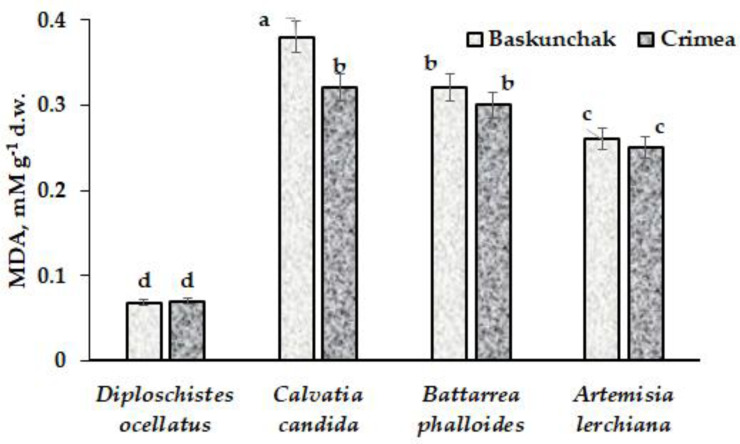
Malonic dialdehyde content in *Diploschistes ocellatus*, *Calvatia candida*, *Battarrea phalloides* and *Artemisia lerchiana* in conditions of Baskunchak Nature Reserve and Crimea. The standard error is shown. Values with the same letters do not differ statistically according to Duncan test at *p* < 0.05.

**Figure 7 plants-12-02530-f007:**
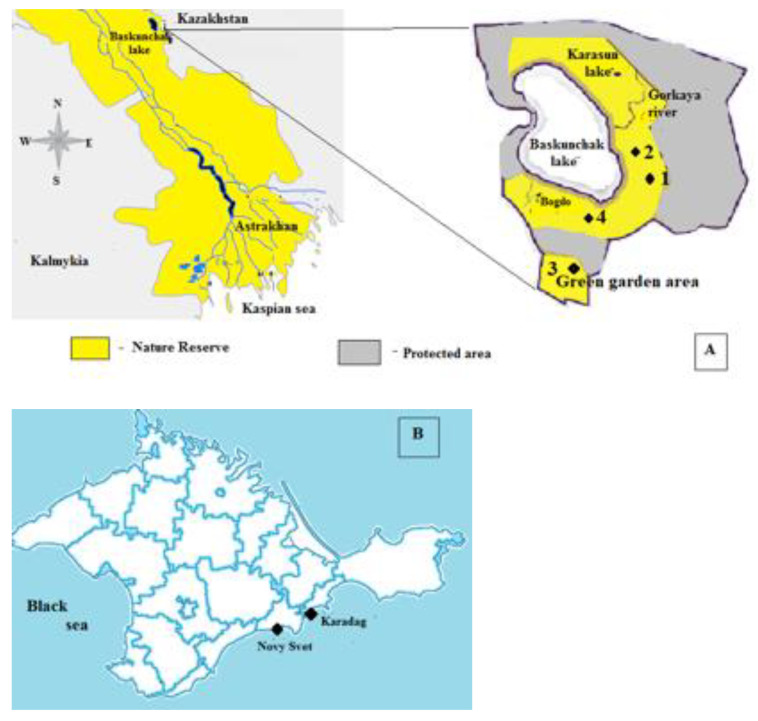
Sampling sites of lichens *Diploschistes ocellatus* (1), mushrooms *Calvatia candida* (Rostk.) Hollós (2) and *Battarrea phalloides* (Dicks.) Pers (3), and wormwood *Artemisia lerchiana* (4)*,* at the territory of Bogdinsko-Baskunchak Nature Reserve (**A**) and at the Crimean Southern shore (**B**).

**Table 1 plants-12-02530-t001:** Biological activity of *Calvatia* species, *Diploschistes ocellatus* and *Artemisia lerchiana*.

	Characteristics	Biological Effect	Biologically Active Compounds
*Calvatia* species	Terrestrial saprophytes	Antibacterial, antiviral, antitumor and wound healing properties [9]	Calvatin and calvatic acid
*Diploschistes ocellatus*	Symbiotic organism of fungi and algae *Trebouxia*, calcephitis [10]	Photo-protective, anti-microbial to gram-positive and gram-negative bacteria, anti-fungal to *Enterococcus faecalis* [11]	Polyphenols, flavonoids
*Artemisia lerchiana*	Haloxerophitis with high tolerance to water deficiency and soil salinity	Allopathic, fungicidal, bactericidal and antioxidant [4]	Essential oil (camphor derivatives)

**Table 2 plants-12-02530-t002:** Mineral composition of *Diploschistes ocellatus*, *Calvatia candida*, *Battarrea phalloides* and *Artemisia lerchiana* at Baskunchak Nature Reserve and Karadag Nature Reserve (mg Kg^−1^ d.w.).

Element	*Diplishistes ocellatus*		*Battarrea phalloides*		*Calvatia candida*		*Artemisia lerchiana*	
	*Baskunchak*	*Crimea*	*Baskunchak*	*Crimea*	*Baskunchak*	*Crimea*	*Baskunchak*	*Crimea*
Ca	114,617 ± 11,210 a	100,656 ± 10,006 a	10,014 ± 1000 c	1020 ± 100 e	3596 ± 317 d	850 ± 82 e	9728 ± 934 c	13,600 ± 133 b
K	1389 ± 129 e	987 ± 96 f	8044 ± 776 d	34,406 ± 3101 a	22,543 ± 2052 c	26,754 ± 2588 bc	22,306 ± 2180 c	28,195 ± 2800 b
Mg	934 ± 93 d	352 ± 32 f	3052 ± 298 a	952 ± 93 d	1514 ± 148 c	672 ± 67 e	1815 ± 180 bc	1858 ± 184 b
Na	873 ± 83 c	831 ± 80 c	328 ± 33 e	502 ± 47 d	10,851 ± 1001 a	11,080 ± 1000 a	3357 ± 328 b	2939 ± 289 b
P	302 ± 27 d	373 ± 33 d	4366 ± 431 b	8167 ± 802 a	4017 ± 389 b	9655 ± 925 a	1938 ± 190 c	2175 ± 207 c
B	7.90 ± 0.68 c	7.24 ± 0.69 c	23.65 ± 2.20 b	6.59 ± 0.62 c	4.10 ± 0.40 d	2.00 ± 0.20 e	22.23 ± 2.11 b	46.99 ± 4.7 a
Co	1.23 ± 0.12 a	0.28 ± 0.03 c	1.04 ± 0.11 a	0.20 ± 0.02 d	0.07 ± 0.01 e	0.02 ± 0.01 f	0.25 ± 0.02 c	0.64 ± 0.06 b
Cu	5.18 ± 0.49 d	2.80 ± 0.30 e	82.14 ± 8.01 a	103.00 ± 10.10 a	9.96 ± 0.90 c	11.21 ± 1.10 c	10.17 ± 1.01 c	19.64 ± 1.88 b
Fe	5072 ± 489 a	1347 ± 127 b	837 ± 82 c	267 ± 25 e	145 ± 14.2 f	109 ± 10 g	444 ± 44 d	1542 ± 145 b
I	15.05 ± 1.45 a	1.23 ± 0.11 bc	0.76 ± 0.07 d	0.40 ± 0.04 e	0.40 ± 0.04 e	0.33 ± 0.03 e	1.03 ± 0.09 c	1.34 ± 0.12 b
Li	3.50 ± 0.34 a	0.96 ± 0.09 c	0.65 ± 0.06 e	0.49 ± 0.05 f	0.08 ± 0.01 g	0.07 ± 0.01 g	0.89 ± 0.08 d	2.36 ± 0.20 b
Mn	29.20 ± 2.88 c	14.00 ± 1.30 d	62.46 ± 6.12 b	14.50 ± 1.43 d	9.09 ± 0.90 e	6.59 ± 0.60 f	57.48 ± 5.71 b	117 ± 10 a
Mo	0.31 ± 0.03 b	0.23 ± 0.02 c	0.93 ± 0.09 a	0.15 ± 0.01 d	0.15 ± 0.01 d	0.12 ± 0.01 f	0.31 ± 0.03 b	0.98 ± 0.09 a
Se	0.25 ± 0.02 e	0.15 ± 0.01 f	6.14 ± 0.59 b	8.13 ± 0.80 a	0.59 ± 0.06 d	0.85 ± 0.08 c	0.08 ± 0.01 g	0.29 ± 0.03 e
Si	22.26 ± 2.01 c	15.06 ± 1.43 c	5.93 ± 0.60 f	54.90 ± 5.20 a	5.68 ± 0.57 f	38.67 ± 3.82 b	10.06 ± 1.00 e	13.42 ± 1.29 d
Zn	16.08 ± 1.55 g	30.24 ± 3.00 ef	107.00 ± 10.00 a	59.30 ± 5.35 b	39.19 ± 3.85 cd	28.12 ± 2.80 f	33.19 ± 3.30 de	44.91 ± 4.38 c
Al	3139 ± 301 a	1086 ± 100 b	193 ± 18 e	126 ± 12 f	99 ± 9 g	59 ± 6 h	346 ± 32 d	721 ± 72 c
As	1.41 ± 0.11 a	1.48 ± 0.15 a	0.90 ± 0.09 b	0.28 ± 0.02 d	0.04 ± 0.01 d	0.02 ± 0.01 f	0.20 ± 0.02 e	0.70 ± 0.07 c
Cd	0.06 ± 0.01 f	0.30 ± 0.03 d	1.20 ± 0.10 a	0.63 ± 0.06 c	0.84 ± 0.08 b	0.53 ± 0.05 c	0.12 ± 0.01 e	0.08 ± 0.01 f
Cr	54.46 ± 5.10 a	2.33 ± 0.21 c	3.39 ± 0.34 b	0.69 ± 0.06 e	0.67 ± 0.06 e	0.28 ± 0.03 f	1.10 ± 0.10 d	1.98 ± 0.20 c
Ni	10.90 ± 1.92 a	3.35 ± 0.33 bc	3.77 ± 0.35 b	0.51 ± 0.05 d	0.30 ± 0.03 b	0.12 ± 0.01 e	3.03 ± 0.30 c	3.04 ± 0.30 c
Pb	5.54 ± 0.52 b	15.48 ± 1.47 a	1.09 ± 0.10 c	0.70 ± 0.07 d	0.10 ± 0.01 e	0.07 ± 0.01 f	0.62 ± 0.06 d	1.09 ± 0.11 c
Sr	114.00 ± 11.00 a	49.09 ± 4.78 b	56.00 ± 5.20 b	4.67 ± 0.05 c	2.79 ± 0.30 d	1.06 ± 0.10 e	48.14 ± 4.80 b	107.00 ± 10.00 a
V	6.78 ± 0.63 b	14.68 ± 1.44 a	2.89 ± 0.30 c	0.51 ± 0.04 d	0.31 ± 0.03 e	0.29 ± 0.03 e	2.69 ± 0.24 c	6.59 ± 0.55 b

Along each line, the values with the same letters do not differ statistically according to Duncan test at *p* < 0.05.

**Table 3 plants-12-02530-t003:** Antioxidant status of *Diploschistes ocellatus*, *Calvatia candida*, *Battarrea phalloides* and *Artemisia lerchiana*.

Object	Location	AOA (mg GAE g^−1^ d.w.)	TP (mg GAE g^−1^ d.w.)	Se (µg kg^−1^ d.w.)	Proline (mg g^−1^ d.w.)
*Diploschistes ocellatus*	Baskunchak	19.0 ± 1.6 b	11.0 ± 1.0 b	250 ± 20 d	1.03 ± 0.10 c
	Crimea *	18.2 ± 1.6 b	10.1 ± 1.0 b	261 ± 21 d	1.02 ± 0.10 c
*Calvatia candida*	Baskunchak	21.1 ± 1.9 b	12.1 ± 1.0 b	590 ± 43 c	1.31 ± 0.12 b
	Crimea **	21.4 ± 2.0 b	11.7 ± 1.0 b	850 ± 78 b	1.22 ± 0.11 b
*Battarrea phalloides*	Baskunchak	7.9 ± 0.7 c	7.4 ± 0.7 c	6140 ± 546 a	1.12 ± 0.10 bc
	Crimea **	7.2 ± 0.7 c	6.9 ± 0.6 c	8130 ± 750 a	1.09 ± 0.10 bc
*Artemisia lerchiana*	Baskunchak	68.6 ± 5.1 a	21.0 ± 1.8 a	135 ± 11 e	5.45 ± 0.44 a
	Crimea **	58.5 ± 5.2 a	24.1 ± 2.1 a	116 ± 10.0 e	5.29 ± 0.41 a

* Novy Svet, ** Karadag Nature Reserve. AOA—total antioxidant activity; TP—total polyphenols. Within each column, the values with the same letters do not differ statistically according to Duncan text at *p* < 0.05.

**Table 4 plants-12-02530-t004:** Photosynthetic pigments content in *Diploschistes ocellatus* and *Artemisia lerchiana*.

Parameter	*Diploschistes ocellatus*	*Artemisia lerchiana*
Chlorophyll a (mg 100 g^−1^ d.w.)	6.18 ± 0.53 b	41.70 ± 4.01 a
Chlorophyll b (mg 100 g^−1^ d.w.)	13.80 ± 0.70 b	91.00 ± 9.00 a
Total chlorophyll (mg 100 g^−1^ d.w.)	19.98 ± 1.40 b	132.70 ± 1.24 a
Chlorophyll b/Chlorophyll a ratio	2.23 ± 0.20 a	2.18 ± 0.20 a
Carotene (mg 100 g^−1^ d.w.)	0.65 ± 0.05 b	19.00 ± 1.6 a
Total chlorophyll/carotene ratio	30.74 ± 3.00 a	6.98 ± 0.70 b

Along each line, the values with the same letters do not differ statistically according to Duncan test at *p* < 0.05.

## Data Availability

Not applicable.

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
