# Peer review of "Comparative Evaluation of Antioxidant Status and Mineral Composition of Diploschistes ocellatus, Calvatia candida (rostk.) Hollós, Battarrea phalloides and Artemisia lerchiana in Conditions of High Soil Salinity"

_plants, 2023, doi:10.3390/plants12132530_

Round 1
Reviewer 1 Report
In the manuscript “
Comparative evaluation of antioxidant status and mineral composition of Diploschistes ocellatus, Calvatia candida (rostk.) Hollós, Battarrea Phalloides and Artemisia Lerchiana in conditions of high soil salinity”, authors Nadezhda Golubkina, Tatiana Ju. Tolpysheva, Vladimir Lapchenko, Helene Lapchenko, Nikolay Pirogov, Viacheslav Zaitsev, Agnieszka Sekara, Alessio Tallarita, Vasile Stoleru, Otilia Cristina Murariu and Gianluca Caruso evaluatied mineral and natural antioxidant accumulation in wormwood Artemisia lerchiana, lichen Diploschistes ocellatus and mushrooms Calvatia candida and Battarrea phalloides, in the conditions of Bogdinsko-Baskunchak Nature Reserve.
Abstract
Please write in the abstract the whole word for GAE.
OK
Introduction
P 2 L 57-58 Gypsum outcrops, intense winds bringing dust and salt are additional factors in ducing significant oxidative stress at the Reserve territory. Stress for whom?
P 2 L 63-65 The following statement is not clear: In this respect, the comparison of the characteristics related to three systematic groups of flora greatly differing by metabolism intensity, life span duration and nutrition, i.e., lichens, mushrooms and vascular plants, is expectedly
P 2 L 65 interesting, Among the …. Write full stop instead of a coma
Materials and methods
P 13 L 447 Proline concentration was determined according to [46], with slight modifications. Write the name of the author
Methods are suitable, adequately described, and used in a way that is possible to replicate.
Results and Discussion
The first sentence in the "Results and Discussion" section is not appropriate. I suggest you write a sentence that introduces the reader to the article's content.
Table 1 and 3 . Biological activity of Diploschistes ocellatus and Artemisia lerchiana. Please, write Diploschistes ocellatus and Artemisia lerchiana in italic.
P 3 L 108 – 110 The following statement is not clear: Data presented in Table 2 and Figure 1 are the first detailed mineral composition of Diploshistes ocellatus, Calvatia candida, Battarrea phalloides and Artemisia lerchiana belonging to three systematic groups: lichens, mushrooms and plants.
Table 2. Mineral composition of Diploshistes esculatus, Calvatia candida, Battarrea ohalloides and Ar-117 temisia lerchiana at Baskunchak Nature Reserve and Karadag Nature Reserve. Write clearly why you compare the mineral composition of the plants from Baskunchak Nature Reserve and Karadag Nature Reserve.
P4 L 132 Nature Reserve. . Delete one full stop
P5 L 140-141 Begin the statement differently, please. Table 2 data indicate that Ca content in A. lerchiana, Calvatia candida and Battarrea 140 phalloides are 11.8 and 31.9 times lower, respectively, than Ca level in lichen.
P 6 L 183 [23.24]… Write coma instead of a full stop
P 7 L 229 vascular plants. is pre… delete full stop
P 8 L 289 A. Lerchiana polyphenol content was double…Write Lerchiana with a small letter
The results are well presented, figures and tables are correct.
Conclusion:
OK.
Specific comments
The introduction is informative, precise, and comprised of relevant content. The manuscript presents new findings and has high novelty.
It is no ethical problem involved.
My suggestion: minor revision
English language is OK.
Author Response
Dear Reviewer,
thank you very much for your valuable comments. We have revised the manuscript according to your recommendations and have highlighted all amendments/modifications by red color.
Answers to the Reviewer’s comments:
1)Abstract
Please write in the abstract the whole word for GAE.
OK
Answer: We have revised the Abstract and reduced the number of words to 250.
Introduction
2) P 2 L 57-58 Gypsum outcrops, intense winds bringing dust and salt are additional factors inducing significant oxidative stress at the Reserve territory. Stress for whom?
Answer: we have modified the sentence to ‘all Reserve flora and fauna species'.
3)P 2 L 63-65 The following statement is not clear: In this respect, the comparison of the characteristics related to three systematic groups of flora greatly differing by metabolism intensity, life span duration and nutrition, i.e., lichens, mushrooms and vascular plants, is expectedly
Answer: the sentence has been changed to: ‘In this respect, the comparison of the mineral composition and antioxidant status related to three systematic groups…’.
4)P 2 L 65 interesting, Among the …. Write full stop instead of a coma
Answer: addressed.
5)Materials and methods
P 13 L 447 Proline concentration was determined according to [46], with slight modifications. Write the name of the author
Answer: addressed.
Methods are suitable, adequately described, and used in a way that is possible to replicate.
Results and Discussion
6)The first sentence in the "Results and Discussion" section is not appropriate. I suggest you write a sentence that introduces the reader to the article's content.
Answer: we have added the required information (lines 88-95).
7)Table 1 and 3 . Biological activity of Diploschistes ocellatus and Artemisia lerchiana. Please, write Diploschistes ocellatus and Artemisia lerchiana in italic.
Answer: addressed.
8)P 3 L 108 – 110 The following statement is not clear: Data presented in Table 2 and Figure 1 are the first detailed mineral composition of Diploshistes ocellatus, Calvatia candida, Battarrea phalloides and Artemisia lerchiana belonging to three systematic groups: lichens, mushrooms and plants.
Answer: we have revised the sentence, changing it to ‘In this respect, mineral composition of Diploshistes ocellatus, Calvatia candida, Battarrea phalloides and Artemisia lerchiana represents a valuable parameter to compare lichens, mushrooms and plants in terms of adaptability to stress conditions. (lines 112-114).
9)Table 2. Mineral composition of Diploshistes esculatus, Calvatia candida, Battarrea ohalloides and Ar-117 temisia lerchiana at Baskunchak Nature Reserve and Karadag Nature Reserve. Write clearly why you compare the mineral composition of the plants from Baskunchak Nature Reserve and Karadag Nature Reserve.
Answer: we have added the required information (lines 88-95).
10)P4 L 132 Nature Reserve. . Delete one full stop
Answer: addressed.
10)P5 L 140-141 Begin the statement differently, please. Table 2 data indicate that Ca content in A. lerchiana, Calvatia candida and Battarrea phalloides are 11.8 and 31.9 times lower, respectively, than Ca level in lichen.
Answer: addressed.
11)P 6 L 183 [23.24]… Write coma instead of a full stop
Answer: addressed.
12)P 7 L 229 vascular plants. is pre… delete full stop
Answer: addressed.
13)P 8 L 289 A. Lerchiana polyphenol content was double…Write Lerchiana with a small letter
Answer: addressed.
Reviewer 2 Report
The manuscript presents an interesting comparison of antioxidant properties and mineral composition among different species isolated from high soil salinity. This type is unsual and it important to studythe interactions among species. In my opinion the manuscript is OK for publication.
I suggest
1. improvement in the quality of figures
2. In material and methods it was described that the phenolics were extracted at 80o C. Please, explain the high temperature used.
Author Response
Dear Reviewer,
thank you very much for your valuable comments. We have revised the manuscript according to your recommendations and have highlighted all amendments/modifications by red color.
Answers to the Reviewer’s comments:
- improvement in the quality of figures
Answer: all the figures have been revised according to the recommendations.
- In material and methods it was described that the phenolics were extracted at 80o C. Please, explain the high temperature used.
Answer: we have inserted the following answer inside the text (lines 278-279): ‘The efficiency of this extraction is determined by short duration, quick deactivation of enzymes at 80 оС, which prevents polyphenols destruction, stable at high temperature [43]’.
Reviewer 3 Report
The current study evaluated mineral composition and antioxidant status of three systematic groups of organisms. Much more data was showed to support current study.
Major concerns:
(1) The writing should be further improved throughout the manuscript. For example, the last three paragraphs should be combined into one paragraph. Too many two or three sentences were used as an independent paragraph.
(2) Figures and tables were made in nonstandard form. The “three-line table” means only three lines in one table, and please delete extra lines.
(3) The size of figures were not uniform in one figure such as Figure 1 including three figures (A, B, and C), but the size of these three figures were not the same!
(4) Please add ±standard deviation in all tables.
The quality of English is fine.
Author Response
Dear Reviewer,
thank you very much for your valuable comments. We have revised the manuscript according to your recommendations and have highlighted all amendments/modifications by red color.
Answers to the reviewer comments.
(1) The writing should be further improved throughout the manuscript.
For example, the last three paragraphs should be combined into one
paragraph. Too many two or three sentences were used as an independent
paragraph.
Answer: addressed.
(2) Figures and tables were made in nonstandard form. The “three-line
table” means only three lines in one table, and please delete extra lines.
Answer: addressed.
(3) The size of figures were not uniform in one figure such as Figure
1 including three figures (A, B, and C), but the size of these three
figures were not the same!
Answer: addressed.
(4) Please add ±standard deviation in all tables.
Answer: addressed.
Round 2
Reviewer 3 Report
Please check all tables and figures which are still not good enough for publication.
Author Response
Dear Reviewer,
thank you very much for your valuable comments. We have checked and revised all Tables and Figures according to your recommendations and have highlighted all amendments/modifications by red color